# Facile Synthesis of Magnetic *Nigella Sativa* Seeds: Advances on Nano-Formulation Approaches for Delivering Antioxidants and Their Antifungal Activity against *Candida albicans*

**DOI:** 10.3390/pharmaceutics15020642

**Published:** 2023-02-14

**Authors:** Maqsood Ahmad Malik, Laila AlHarbi, Arshid Nabi, Khalid Ahmed Alzahrani, Katabathini Narasimharao, Majid Rasool Kamli

**Affiliations:** 1Chemistry Department, Faculty of Science, King Abdulaziz University, P.O. Box 80203, Jeddah 21589, Saudi Arabia; 2Department of Chemistry, University of Malaya, Kuala Lumpur 50603, Malaysia; 3Department of Biological Sciences, Faculty of Sciences, King Abdulaziz University, P.O. Box 80203, Jeddah 21589, Saudi Arabia; 4Center of Excellence in Bionanoscience Research, King Abdulaziz University, P.O. Box 80203, Jeddah 21589, Saudi Arabia

**Keywords:** bionanomaterial, carbon framework, *Nigella Sativa*, magnetic seeds, biomedical, antioxidant

## Abstract

This article reports on incorporating magnetic nanoparticles into natural carbon frameworks derived from *Nigella Sativa* seeds and their synthesis via co-precipitation reactions for application in biomedicine. The magnetic *Nigella Sativa* Seeds (Magnetic NSS), a metal oxide-based bio-nanomaterial, has shown excellent water diaper presence due to the presence of a wide range of oxygenous hydroxyl and carboxyl groups. The physicochemical properties of the composites were characterized extensively using Fourier transform infrared spectroscopy (FTIR), powder-X-ray diffraction (XRD), scanning electron microscopy (SEM), elemental analysis, transmission electron microscopy (TEM), and vibrating-sample magnetometer. Furthermore, synthesized magnetic NSS showed antioxidant and antifungal activity. The antifungal susceptibility was further tested against *Candida albicans* with a MIC value of 3.125 µg/mL. Analysis of antioxidant defense enzymes was determined quantitatively; the results suggested that antioxidant enzyme activity increase with increased magnetic NSS concentration. Furthermore, biofilm inhibition assay from scanning electron microscopy results revealed that magnetic NSS at the concentration of 3.5 μg/mL has anti-biofilm properties and can disrupt membrane integrity.

## 1. Introduction

Bioinspired nanomaterials or bionanomaterials (NMs) are emerging as the most promising areas of green chemistry research concerning science and engineering [1]. These materials provide a unique scholarly platform for discussing and reporting the sensitive functional properties of materials structured by natural resources. These materials can be produced by bio-templating nanoparticles (NPs) in natural organic frameworks [2]. A new area of bionanomaterial creation has been introduced to develop carbon frameworks derived from wild plants with unique structural and functional properties [3]. NPs are proven to the effective therapeutic tools and are involved in a wide range of pharmaceutical applications, including tissue engineering, drug delivery, and tumor detection. Due to their diminutive size, these particles can quickly enter and translocate within the cells. In addition, the physicochemical characteristics, shape, size, and surface of NPs can affect their cellular absorption, targeting, and cytotoxicity [4].

Iron oxides and hydroxides can be handled using a low-gradient external magnetic field. Magnetic quality, surface defects, high surface area-to-volume ratio, and high intrinsic reactivity of surface sites of iron oxide particles are helpful in targeted activities [5]. Hence, magnetic particle impregnation in abundant, nontoxic, having rich functional groups can be the foremost and wise approach for biomedical application [6]. Most processes currently used for producing these target-oriented NMs are costly, dependent on non-ecological materials, and generate harmful waste. The development of nanotechnology based on principles of green chemistry is a rational approach that embraces important aspects of the development of an ecologically sustainable society [7]. Green chemistry emphasizes using renewable chemicals and designs processes that reduce or eliminate the use and production of hazardous substances. The NMs synthesis, based on green chemistry, involves using natural resources, non-hazardous solvents, and biodegradable and biocompatible materials through energy-dependent processes [8].

A world widely, particularly countries bordering the Mediterranean Sea, Eastern Europe, Middle East, and Western Asia, cultivate *Nigella Sativa* (NSS) seeds (Commonly known to be as Kalaunji), which can act as natural and versatile carbon framework, and possesses various functional and structural properties to assemble various NPs [9]. In addition, NSS can bind to charged molecules/ions because cellulosic surfaces contain carboxylic, hydroxyl, and phenolic functional groups [10]. Therefore, magnet-responsive bionanomaterials based on NSS can be the most well-known technique for the significant applications of materials in the biomedical field.

*Candida* species are the most commonly reported opportunistic fungal pathogen producing mucosal infections, and they are ranked fourth in generating nosocomial infections in humans [11,12]. Among the *Candida* species, *Candida albicans* (*C. albicans*) remain dominant and are responsible for over 90% of invasive infections worldwide [13]. However, *C. albicans* are frequently observed on the oral, gastrointestinal, and vaginal mucosae. In addition to its host-beneficial effects, such as shielding the host from various pathogenic assaults, it has antimicrobial properties [14]. *C. albicans* can form a biofilm and undergo morphological changes to survive in the host microenvironment [15]. *C. albicans* can create polymicrobial biofilms with various bacteria in vitro and in vivo, influencing the disease’s progression and management [16]. Using fluconazole and other antifungal drugs, such as the azole class, has resulted in multidrug-resistant in *C. albicans* [17]. Mutations in the ergosterol production pathway and an increase in antifungal efflux are the two fundamental mechanisms responsible for *C. albicans*’ drug resistance [18]. Previous studies suggest that metal-based NPs have an antifungal activity [19,20]. Due to their distinctive physical and chemical properties, drug delivery, antibacterial properties, and bio-detection are critical features of many metallic NPs [21].

Therefore, the main objective of this study was to utilize the magnetic properties, high surface area-to-volume ratio, and high intrinsic reactivity of surface sites of iron oxide particles for the synthesis of magnetic NSS and their biological applications. Moreover, the present study aimed to study the antifungal effect of magnetic NSS on *C. albicans*. The as-developed magnetic NSS could be used to lower the drug resistance of *C. albicans* with other known antifungal drugs. Moreover, the antioxidant effect of magnetic NSS and its inhibitory actions on biofilm production were also explored. Therefore, this work demonstrates the antifungal efficacy of magnetic NSS against *C. albicans* and suggests that it may serve as an alternative to conventional antifungal therapies.

## 2. Materials and Methods

### 2.1. Chemicals

Ferric chloride hexahydrate (FeCl_3_.6H_2_O) and ferrous sulfate (FeSO_4_) were supplied by Merck, Darmstadt, Germany. Sodium hydroxide (NaOH) and hydrochloric acid (HCl) were supplied by Sigma-Aldrich, Hamburg, Germany. All these chemicals were of analytical reagent grade and used without further purification.

### 2.2. Nigella Sativa Seeds (NSS)

The NSS was used as the carbon template on which magnetic NPs were grown. *Nigella Sativa* seeds were purchased from the local market in Jeddah.

### 2.3. Instrumentation

Fourier transform infrared (FT-IR) spectra (4000–400 cm^−1^) were obtained on VERTEX 70/70 v spectrophotometer (BRUKER, Biopolis St, Singapore) after the formation of pallets in KBr. X-ray diffraction (XRD) pattern of the prepared material was recorded on Philips PW-3710, diffractometer using Cu-Kα radiation (λ = 1.54 Å), a Cu-filter, generator voltage 35 kV, 30 mA current, and a proportional counter detector. The morphology and structure of the prepared material were analyzed by field emission scanning electron microscopy (FESEM (Nova Nano SEM 450, FEI, Hillsboro, OR, USA) equipped with EDAX (Bruker 127 eV) and transmission electron microscopy (TEM) using Tecnai T-30, Hillsboro, OR, USA (300 kV FEGTEM) operating at 80 kV. The magnetic property of magnetic NSS was verified from the measurement at 27 °C under a magnetic field (in the range of 2.2 and −2.2 KOe) using an MPMS-XL-7 magnetometer (Quantum Design, Darmstadt, Germany).

### 2.4. Preparation of Magnetic NSS

NSS was washed and dried overnight at 65 °C in a hot air oven and ground to powder. Then, 0.5 g of NSS powder was dispersed in 50 mL of distilled water using an ultrasonicator. Afterward, 100 mL each of 0.1 M FeCl_3_ and 0.05 M FeSO_4_ solutions were added to the dispersed suspension of NSS and stirred for 60 min on a magnetic stirrer at 65 °C and 800 RPM. After that, 2 M NaOH solution was added dropwise to the above mixture of iron salts and NSS while on continued stirring conditions to maintain higher pH (approximately 10–11). Then, the mixture solution turned into a blackish-brown precipitate and was stirred for 20 min. The obtained precipitate-like particles settled down on stopping the reaction and stayed out for half an hour to cool down, and then the prepared solid was filtered and washed with distilled water several times. Finally, the prepared material was isolated from the aqueous solution using an external magnetic field and dried in a vacuum oven at 50 °C for two days [22].

### 2.5. Determination of In Vitro Antioxidant Activity

Three free radical-scavenging tests were used to determine the in vitro antioxidant activity of magnetic NSSs. All assays for scavenging free radicals were combined with an assessment of the common antioxidant L-ascorbate.

#### 2.5.1. Radical Scavenging Activity Using DPPH

The Blois method was used to evaluate the magnetic NSS and the common antioxidant L-ascorbate for their ability to scavenge free radicals [23]. In separate tubes, 40 microliters of working stock containing conventional antioxidants and 0.01–0.1 mg (10–100 g) of magnetic NSS were taken. The reaction mixture’s volume was changed to 2.0 mL by adding methanol. After that, 150 L of a 0.1 mM metabolic solution of DPPH was added to each tube, and they were all vortexed together. For 20 min, the tubes were held at 25 °C. A combination of 40 L DMSO and 1.960 mL methanol was utilized for the baseline optical density correction on the control, which was made without magnetic NSS. At 517 nm, the samples’ absorbance was gauged. According to the given formula, the proportion of free radical scavenging activity was expressed as a percent inhibition.
% Inhibition DPPH radical = [(absorbance_control_ − absorbance_sample_)/absorbance_control_] × 100

The inhibition concentration, or IC50, is the magnetic NSS needed to cause a 50% reduction in optical density concerning control [24].

#### 2.5.2. Nitric Oxide Radical Scavenging Assay

The Griess Illosvoy reaction was used to gauge the amount of nitric oxide (NO) scavenging activity [25,26]. NO generation happens when sodium nitroprusside breaks down in an aqueous solution at a physiological pH of 7.2. When NO and oxygen interact in an aerobic environment, stable compounds are created (nitrate and nitrite). These products are quantified using Griess reagent. Competition between oxygen and nitric oxide scavengers reduces the formation of nitrite ions. For the experiment, 1.0 mL of phosphate-buffered saline (10 mM at pH 7.4) containing 10 mM sodium nitroprusside was combined with 40 L of DMSO containing 0.01–0.1 mg of magnetic NSS, and the standard antioxidant was added separately and vortexed. For two hours, this reaction mixture was incubated at 30 °C. After incubation, each reaction mixture received 0.5 mL of Griess reagent (1% sulfanilamide, 2% H_3_PO_4_, and 0.1% N-(1-naphthyl) ethylenediamine dihydrochloride). At 550 nm, a chromophore was detected by diazotizing nitrite with sulphanilamide and attaching it to naphthyl ethylenediamine dihydrochloride. The standard antioxidant ascorbic acid and magnetic NSS’s inhibition of nitrite production were measured to the control. The following equation was used to determine the percentage inhibition of the production of nitric oxide radicals:% Inhibition= [(A_0_ − A_1_)/A_0_] × 100

A1 denotes the absorbance in the presence of both magnetic NSS and the standard, and Ao represents the absorbance of the control (in magnetic NSS). The amount of sample (μg or mg) needed to scavenge 50% of the nitric oxide free radical was determined using the inhibition curve [27].

#### 2.5.3. Total Reduction Capability

The approach described by Oyaizu [28] was used to determine the decreasing capacity of magnetic NSS. 2.5 mL of 0.2 M phosphate buffer (pH 6.6) containing (1% *w*/*v*) K_3_Fe(CN)_6_ was added separately to 40 L of DMSO with or without magnetic NSS and standard antioxidant from 0.01–0.1 mg concentrations. For 20 min, this reaction mixture was incubated at 50 °C. 2.5 mL of (10% *w*/*v*) TCA was added after incubation. The 2.5 mL of supernatant was removed from each tube separately after the reaction mixture had been centrifuged for 10 min at 3000 rpm. The color generated was measured at 700 nm after adding 2.5 mL of distilled water and 0.5 mL of FeCl_3_ (0.1%, *w*/*v*) to these supernatants [28]. The lowering power of magnetic NSS was expressed as a rise in absorbance value.

### 2.6. Antifungal Activity

In this study, the antifungal activity of magnetic NSS was evaluated against *C. albicans* SC5314 (laboratory control strain) from ATCC (American Type Culture Collection). The strain was stored at −80 °C supplemented with 20% glycerol. For experimental purposes, the strain from glycerol stock was recovered on Sabouraud Dextrose (SDA) agar plates at 37 °C for 24 h.

The antifungal activity of magnetic NSS against a *C. albicans* isolate was assessed using a broth microdilution test following the Clinical & Laboratory Standards Institute (CLSI-2008, www.clsimena.org; accessed on 22 September 2021) recommendations. The magnetic NSS was mixed with 1% DMSO (Sigma-Aldrich Co., St. Joseph, MI, USA) and tested in a 100 to 0.195 g/mL concentration range. Two-fold dilutions of test compound (100 μL) were prepared in 96-well flat-bottom microtiter plates, inoculated with 100 μL of inoculum (1 × 10^6^ cells/mL), and incubated at 37 °C for 24 h. Positive and negative controls were utilized as fluconazole and 1% DMSO, respectively. Post-incubation, MICs for magnetic NSS and fluconazole were determined visibly as the lowest concentrations that inhibited the growth of *C. albicans*. The experiments were repeated thrice to validate the results.

#### 2.6.1. Effect of Magnetic NSS on Antioxidant Enzymes in *C. albicans*

The effect of magnetic NSS on antioxidant enzymes such as catalase, superoxide dismutase, glutathione peroxidase, glutathione reductase, and glutathione transferase was studied in *C. albicans*. The antioxidant activity of these enzymes was checked, as reported by [19]. In brief, *C. albicans* SC5314 cells were treated with various doses of magnetic NSS (0.5 MIC and MIC) for 4 h. The exposed cells were subsequently employed to prepare cell-free extracts (CFE). The supernatant was used to measure the activity of antioxidant enzymes, whereas the pellet was utilized to estimate the degree of lipid peroxidation (LPO).

#### 2.6.2. Catalase (CAT)

The CAT activity was measured by combining CFE, H_2_O_2_ (30% *w*/*w*), and PBS. Using a Shimadzu UV-1800 spectrophotometer, the consumption of H_2_O_2_ in the reaction mixture was determined every 30 s for 3 min at 230 nm. Utilizing an extinction coefficient of 0.081 × 10^−1^/mM/cm, the CAT activity was estimated as H_2_O_2_ (mol) consumed per minute.

#### 2.6.3. Superoxide Dismutase (SOD)

CFE was combined with Trizima buffer (50 mM; pH 8.5) and pyrogallol to test SOD activity (20 mM; Sigma-Aldrich, USA). The spectrophotometer Shimadzu UV-1800 was used to quantify the activity as the suppression of pyrogallol autoxidation at 420 nm.

#### 2.6.4. Glutathione Peroxidase (GPx)

The GPx activity was determined by measuring the change in absorbance caused by NADPH oxidation. The measurements were taken at 340 nm, and the activity was calculated as µmol NADPH oxidized per minute with an extinction coefficient of 6.22 × 10^3^/M/cm.

#### 2.6.5. Glutathione Reductase (GLR)

The GLR activity was calculated using the rate of NADPH oxidation, and measurements were made at 340 nm. The enzyme activity was calculated as µmol NADPH oxidized per minute using an extinction coefficient of 6.22 × 10^3^/M/cm.

#### 2.6.6. Glutathione Transferase (GST)

GST activity was measured spectrophotometrically at 340 nm to determine the rate of GSH-CDNB conjugation. The enzyme activity was measured in 1 µmol of glutathione conjugated per minute.

#### 2.6.7. Lipid Peroxidation (LPO)

LPO was estimated using thiobarbituric acid reactive substances (TBARS). CFE pellets were suspended in pH 7 Tris-HCl buffer (Tocris Biosciences, UK). Solubilized pellet (200 μL) was mixed with PBS (1.8 mL; pH, 7) and incubated for 1 h at 37 °C in a water-bath shaker. TCA (10%) and TBA (0.67%) were added to the tubes to stop the reaction. Cooled tubes were centrifuged at 2500 g for 10 min after 20 min in boiling water. Finally, TBARS absorbance was measured at 432 nm, and the extinction coefficient, 1.56 × 10^5^/M/cm, was used to calculate the nmol TBARS generated.

### 2.7. Effect on Biofilm Formation in C. albicans

A standardized cell suspension of 200 μL of *C. albicans*, with a 0.5 McFarland Standard, was made, following inoculation into a 96-well flat bottom microtiter plate. The plates were incubated at 37 °C for 2 h. Upon incubation, the medium was aspirated following a two-time rinse with sterile PBS, which allowed for the removal of the planktonic cells. Later, 100 μL of the Magentic NSS was added at varying concentrations to each well, followed by fresh medium (100 μL), and the plates were further incubated at 37 °C for 24 h. The metabolic activity of the biofilms was then observed using the 3-(4,5-dimethyl-2-thiazolyl)-2,5-diphenyl-2H-tetrazolium bromide (MTT) reduction assay. A stock of MTT was formulated in PBS. After the incubation, 50 μL of a 5 mg/mL MTT solution was added to all 96-well flat-bottom microtiter plates, followed by overnight incubation at 37 °C. The MTT was then removed, and wells were washed twice with PBS, followed by the addition of 100 μL of DMSO. The results were read using the multi-well microplate reader (iMark, BioRad) at a wavelength of 490 nm.

Furthermore, crystal violet staining was performed to perform light microscopy analysis. Briefly, upon the incubation, the wells were gently washed with sterile PBS, and a 5% solution of crystal violet was added to each well. The plates were left at room temperature for 10 min to facilitate staining. The stain was then removed, and the wells were again washed with sterile PBS, allowing the access removal of the crystal violet. The glass coverslips were then inverted onto a slide and visualized using a light microscope.

### 2.8. Confocal Laser Scanning Microscopy (CLSM)

Confocal laser scanning microscopy was used to study how magnetic NSS affects *C. albicans* biofilms [19]. *C. albicans* SC5314 was grown on glass coverslips in 6-well microtiter plates for 2 h at 37 °C under ideal biofilm conditions. After the 2 h incubation, the wells were rinsed with sterile PBS to remove planktonic cells, and the magnetic NSS was distributed into the designated wells at MIC and 1/2 MIC concentrations. The plates were incubated at 37 °C for 24 h without magnetic NSS in the growth control and sterility wells. After incubation, the planktonic cells were aspirated after two washes with fluorescent dye FUN-1 (Invitrogen, Thermo Fisher Scientific, RSA), concanavalin A (ConA)-Alexa Fluor 488 conjugate, and PBS. The coverslips were transferred to a new 6-well microtiter plate and incubated for 45 min at 37 °C with 2 mL PBS containing FUN-1 (10 μM) and ConA-Alexa Fluor 488 conjugate (25 μg/mL) to stain. Lightless incubation occurred. The Zeiss Laser Scanning Confocal Microscope (LSM) 780 and Airyscan were used to view the stained biofilms on coverslips inverted glass slides after incubation (Carl Zeiss).

## 3. Results and Discussion

### 3.1. Mechanism Involvement in the Preparation

The preparative reaction mechanism can be understood from the previous research [22,29,30]. Initially, the dispersed cellulosic surface of NSS having numerous functional groups, -OH, -C-O, C=O, N-H, and -COOH (as confirmed from FTIR analysis) electrostatically absorbed the precursors (Fe^2+^ and Fe^3+^) on its surface in the aqueous mixture. After that, the co-precipitation process might have started through the generation of hydroxides of the iron ions that might have led to the nucleation and the growth of the iron oxide NPs on the NSS surface (Figure 1).

It has to be noted that during the process, the high pH (using the high concentration of base (NaOH)) might have resulted in the high contribution of the hydroxyl (OH^−^) ions in the solution. This increases the nucleation rate and leads to the development of tiny nuclei meaning nano-size particles grown on the NSS surface [30]. Of note, the reaction occurred at 65 °C; therefore, NSS might have released the extract having the phytochemicals. These phytochemicals might have also worked as stabilizing and capping agents, thus given the nanosized particles.

### 3.2. Synthesis and Characterization of Magnetic NSS

The magnetic NSS was characterized based on many techniques, as mentioned below. The FTIR spectrum of bare iron oxide NPs prepared through the co-precipitation method showed the absorption bands for metal-oxygen (Fe-O) at 634 and 453 cm^−1^, along with the vibrational bands of the hydroxyl group at 3405 and 1623 cm^−1^ (due to the moisture absorbed on to the NPs surface), respectively. According to some previous research [31,32], the peaks observed in the FT-IR spectrum of NSS powder confirm the presence of many functional groups on the cellulosic surface of NSS. Therefore, the pre-assumption for this study was that a cellulosic surface of NSS having a large number of carboxylic (-COOH) and phenolic (-OH) groups could be acted as an interactive carbon framework to accommodate Fe^3+^ and Fe^2+^ ions to form Fe^2+^/Fe^3+^ impregnated NSS [9]. In addition to NaOH, these ions underwent oxidation via co-precipitation, which produced magnetic NPs on the NSS carbon framework [22]. The prepared product was referred to as magnetic NSS due to M-O bonding between the oxygenous group on NSS and iron ions in the solution (Figure 1) [22]. This mechanism can be confirmed from the FTIR spectrum of magnetic NSS.

The FT-IR spectrum of the NSS framework after incorporation of magnetic NPs gave characteristics peaks for confirming the preparation of hybrid composite, magnetic NSS. The FT-IR absorption peaks in Figure 2 for magnetic NSS, appearing in the range 3400–800 cm^−1^, confirmed the presence of various functional groups in the cellulosic framework of NSS. A broad peak around 3393 cm^−1^ was assigned for O-H stretching, and two strong peaks around 2926 and 2865 cm^−1^ appeared for -C-H stretching of -CH_3_ and -CH_2_ groups, respectively. The -C=C- stretching frequency was assigned to peak at 1393 cm^−1^, whereas acidic (-COOH) and ketonic groups (-C=O) were confirmed from the peaks at 1351 and 1744 cm^−1^, respectively. The IR band around 1118 cm^−1^ appeared for -C-H deformation of the alkene group or may be due to the C-O stretching. The presence of amino acids in the seeds was confirmed from the peak around 1604 cm^−1^, which was attributed to amide bonds of the protein–peptide bond. The peaks between ~900–800 cm^−1^ were assigned for the C-H vibration. The spectrum of magnetic NSS (Figure 2) showed all the similar peaks that appeared for NSS as reported in various previous research [9,10]. A new band related to the stretching vibration of metal–oxygen (Fe-O) was observed in the range 650–440 cm^−1^, which might be attributed to metal-O vibrations. A metal–oxygen peak was evidence of the interaction between NSS and magnetic NPs.

These results showed good agreement with the previous research [9,10,22]. Scientists from other countries have also worked on NSS and found almost similar peaks to ours, although their variety and species might differ from ours. For instance: El-Bediwi et al. reported the influence of ultraviolet-C (UVC) on the growth behavior and internal structure of the *Nigella Sativa* plant and observed the FTIR spectrum [33]. They analyzed that almost similar peaks were observed for normal NSS and after exposure to UVC for 1, 2, 3, and 4 h with only a slight difference in the distance. Of note, these peaks also match our FTIR, although the variety of our BC seed may be different from them. However, since both types might have the same basic structure and constituents, they give similar peaks. The African Scientists also observed almost identical IR peaks for NSS [34]. The FTIR spectra (ranging 4000–500 cm^−1^) showed the functional groups on the surface of the (pristine NSS, 300 °C, carbonized NSS, 10% H_2_SO_4_ treated NSS (NSS-10) and 20% H_2_SO_4_ treated NSS (NSS-20). NSS-10 and NSS-20 showed almost similar peaks for each other, whereas the other showed a very slight difference with these two. This is because all have similar organic constituents. These FTIR spectrums of NSS showed maximum similarity to the NSS used in this study. Some slight changes might be due to different varieties and species of NSS and conditions (preparation, washing, heating, and others). Further addition of new peaks at 613 and 426 cm^−1^ strongly revealed the growth of inorganic particles (due to Fe-O vibrations) on the NSS surface.

X-ray diffraction pattern of magnetic NSS showed various peaks corresponding to NSS and NPs in the range of 10–80°. The broad peak in the range 20.0–24.0° (2θ) (Figure 3) corresponded to the cellulosic structure of NSS. Many studies based on organic contents such as natural plant seeds have almost identical spectra due to the specific pattern of cellulosic surface. The peak position in XRD is simply a fingerprint of the material (regardless of the size), so one achieves the same peak positions for the same material. The peak position of XRD depends on the perpendicular lattice parameter, which should present at an angle determined by Bragg law. This issue is different in the different materials but similar for the similar material.

The XRD spectrum of a powder should also be the same, either in bulk or nanomaterials, as the chemical composition and bonding are the same regardless of the particle size. Therefore, peaks should keep the same position. This is a very common physio-chemistry of XRD analysis. For natural materials, all the diffraction peaks were indexed to the structure of cellulose in natural plant materials [22]. The observed peaks could be assigned to the reflections from the (002) planes in the order of ascending 2θ values. For instance: El-Bediwi et al. discussed the XRD pattern of normal NSS after exposure to UVC for 4 h [33]. It is reported that almost similar peaks were observed for normal NSS and after exposure to UVC for 1, 2, 3, and 4 h, as in the present study. Some differences in the peak position can be seen only due to the internal stresses in materials. For this study, after the incorporation of inorganic particles, the intensity of the diffraction peaks that appeared for magnetic NSS reduced (as compared with the previous research). This might be due to the interaction between NPs and oxygenous groups available on NSS. Furthermore, the XRD pattern contained characteristics peaks for magnetic NPs (iron oxide) around 30.1, 35.4, 43.7, 53.6, 57.0, and 63.01 (2θ°) corresponded to the (220), (311), (400), (422), (511), and (440) planes of the iron oxide, respectively, which match well with the JCPDS card no. 19–0629 or JCPDS card No. 89–691 of iron oxides NPs [22]. However, it should be noted that the color of magnetic NSS was turned from brownish black into brown during XRD analysis, which might be due to the partial oxidation of Fe_3_O_4_ (magnetite) to the γ-Fe_2_O_3_ (maghemite) during synthesis (washing/drying steps) and phase-transition during analysis process [30]. This may indicate the presence of both Fe_3_O_4_ and γ-Fe_2_O_3_ phases in the prepared magnetic NSS. Due to the high similarity between the XRD patterns of Fe_3_O_4_ and γ-Fe_2_O_3_, it is difficult to recognize both phases in the obtained XRD pattern [30]. Hence, the dominant phase of iron oxide in the magnetic NSS composite sample can be confirmed by the angular value feature of XRD data based on 2θ value of 35.49° (highest peak), corresponding to the (311) plane {Standard 2θ value for the (311) plane: 35.423° for Fe_3_O_4_ relative to 35.631° for γ-Fe_2_O_3_} [30]. Notably, the obtained 2θ value of the (311) plane in this work is in good agreement (close) with the standard 2θ value for Fe_3_O_4_, indicating that Fe_3_O_4_ is the dominant phase in the prepared magnetic NSS. Notably, the XRD peaks evident that no other forms of iron oxide {hematite (α-Fe_2_O_3_), goethite (FeO(OH)), or any iron hydroxides} existed in the prepared magnetic NSS. The crystallite sizes were found to be in the range of ~11.8–12.3 nm using the Scherrer equation to XRD peaks, as shown in Table 1.

The morphological properties of raw NSS and magnetic NSS were further evaluated by SEM and TEM images, as seen in Figure 4a–d. The SEM image of magnetic NSS (Figure 4a) showed high surface roughness (relative to the smooth surface of NSS) loaded with quasi-spherical or irregularly shaped bright particles due to the in-situ incorporation of magnetic NPs on the NSS carbon framework. EDX elemental analysis was depicted in Figure 4b, further confirming the incorporation of iron oxide onto the NSS framework because of the detection of C, N, O, and Fe elements with a weight percentage of 60.19, 3.32, 25.74, and 10.75, respectively. The TEM images of magnetic NSS (Figure 4c,d) show the well-developed small black fine NPs associated with rough NSS surface, suggesting the proper growth of iron oxide NPs on the cellulosic structure of NSS. The cellulosic surface acted as an attractive site for the in-situ incorporation of quasi-spherical-shaped magnetic NPs (diameters of 10–20 nm) onto the NSS surface (Figure 4d) for magnetic NSS). TEM images also revealed the formation of aggregated NPs on the NSS surface due to the strong van der Waals interaction and magnetic behavior of NPs. This could subsequently lead to the low dispersibility and stresses of NPs on the surface of NSS, along with the formation of a rough (non-uniform) distribution surface for magnetic NSS. The HR-TEM image (Figure 4e) of magnetic NSS indicates the lattice spacing (d spacing) at 0.48 nm, which shows close agreement with the lattice spacing of (111) planes of the Fe_3_O_4_ structure.

Thermal analysis was also conducted for this study. Three processes were seen in the magnetic NSS’s TGA curve (Figure 5). In the first step, in the broad temperature range of 120 to 350 °C, the sample lost its volatile compounds, which may be the oil in the seeds and moisture on the composite surface. The sample was found to contain ~41% volatile compounds. The second and third steps appeared between 350 and 500 °C and 500 and 600 °C, which might be due to the pyrolysis of the carbon network of NSS in the composite. These measurements indicated that the total weight loss of composite was about ~55%. The three broad exothermic peaks in the DTG curve supported this explanation (Figure 5).

The magnetic property of magnetic NSS was verified from the magnetization curve measured by a Vibrating-sample magnetometer (Figure 6) at room temperature. The saturation moment of magnetic NSS from the hysteresis loop measured from VSM was ~14.9 emu/g [22]. The loop of magnetic NSS showed a shallow coercive field and the remanence value as evidence for the super magnetic character of NSS at room temperature. The results apprised that magnetic NSS can be effectively guided by a magnet.

### 3.3. In Vitro Antioxidant Assays

Magnetic NSS has shown antioxidant activity due to its capacity to scavenge free radicals. Various assays for free radical scavenging are shown in Figure 7a–c.

#### 3.3.1. DPPH Free Radical Scavenging Activity

As seen in Figure 7a, the addition of magnetic NSS at varying doses (0.01–0.1 mg) resulted in a dose-dependent reduction in the activity of DPPH radicals. At a different concentration that can turn the stable DPPH radical (purple) into its non-radical form, DPPH, the free radical scavenging activity of magnetic NSS on DPPH radicals was measured (yellow). When the concentration was increased further, the magnetic NSS radical-scavenging activity increased from 0.01 to 0.05 mg and remained essentially stable. The estimated radical inhibition by L-Ascorbate (L-AA) and magnetic NSS were determined to be 95.0% and 72.0%, respectively, at 0.1 mg concentration. The results also showed that for each concentration tested, L-AA revealed a greater scavenging effect than magnetic NSS (Figure 7a). The IC50 values for L-AA and magnetic NSS were found to be 0.02 ± 0.002 mg and 0.026 ± 0.002 mg, respectively.

The antioxidant capacity of synthetic magnetic NSS was assessed using the DPPH free radical. The stable molecule DPPH has been widely used to evaluate antioxidant activity since it can be lowered by removing hydrogen or electrons [34]. Higher concentrations of magnetic NSS can increase the DPPH free radical scavenger activity, according to our data, which indicated the impact of varied magnetic NSS concentrations on DPPH radical antioxidant activity.

#### 3.3.2. Nitric Oxide Radical Scavenging Activity

Figure 7b demonstrates that, in contrast to L-AA, magnetic NSS showed dose-dependent nitric oxide radical-scavenging activity between 0.01 and 0.1 mg with an IC50 value of 0.033 ± 0.002 mg. Additionally, magnetic NSS had a lowering capability almost identical to L-AA and lowered the nitrite level produced by the breakdown of sodium nitroprusside. The antioxidant capabilities of phytochemicals contained in magnetic NSS compete with oxygen to react with nitric oxide, which inhibits the creation of peroxynitrite radicals and may account for the lower quantity of nitrite synthesis.

#### 3.3.3. Total Reduction Capability

The total reduction capability assay was used to evaluate the magnetic NSS’s antioxidant potential. The rise in absorbance value was utilized to quantify the magnetic NSS’s reduction power. Magnetic NSS had an absorbance value of around 0.25 ± 0.01 at 0.01 mg and 2.54 ± 0.1 at 0.1 mg. L-AA had more potent activity than magnetic NSS at all concentrations. However, magnetic NSS was reported to have the reductive capability for all concentrations (0.01–0.1 mg) (Figure 7c). The reducing power of magnetic NSS explains the likely antioxidant function.

### 3.4. Antifungal Activity of Magnetic NSS

The antifungal activity of magnetic NSS, MICs was determined against *C. albicans* SC5314. The MIC values and fluconazole susceptibility of magnetic NSS and fluconazole are reported in Table 2. Following the CLSI interpretive guidelines for in vitro susceptibility testing of *Candida* species, the isolate is fluconazole susceptible to both NSS and fluconazole.

Nanomaterials can be used against multidrug-resistant pathogens to treat infections [35,36]. Furthermore, the MIC values from this study varied from the other findings [19,20,21]; the results are consistent with the literature, where several biogenic-based metallic nanoparticles were found to possess anti-*Candida* action.

### 3.5. Activity on Antioxidant Enzymes in C. albicans

Antioxidant enzyme activity was measured before and after exposure to magnetic NSS. Quantitative estimates of all the observations are displayed in Figure 8 for antioxidant enzymes and LPO. A higher concentration of magnetic NSS was found to boost catalase activity. CAT activity averaged 0.33 mol of H_2_O_2_ per minute in the untreated negative control and 1.10 mol of H_2_O_2_ per minute in the treated positive control. The activity increased after exposure to the magnetic NSS, with values of 0.32 and 5.06 mol of H_2_O_2_ consumed per minute against 0.5 MIC and MIC, respectively (Figure 8a). SOD activity in *C. albicans* was dramatically affected by exposure to magnetic NSS, with dose-dependent increases in enzymatic activity. Measured values at 0.5 MIC and MIC were 0.28 and 0.66 units/mL, whereas those for negative and positive controls were 0.28 and 0.45 units/mL. (Figure 8b). *C. albican*’s GPx activity was also enhanced when treated with magnetic NSS. Amounts of 0.31 mol of NADPH oxidized/min were recorded for the control sample, whereas values of 0.48 and 1.04 mol of NADPH oxidized/min were recorded for the 0.5 MIC and MIC of *C. albicans*-treated magnetic NSS (Figure 8c). When treated with magnetic NSS, enzymatic activity decreased for GST and GLR. While the negative control yielded 9.5 mol CDNB conjugate per minute and the positive control produced 2.1 mol CDNB conjugate per minute, the GST activity in exposed cells was determined to be 9.0 and 4.1 at 0.5 MIC and MIC, respectively. Findings for GLR activity at 0.5 MIC and MIC were 0.21 and 0.08 mol of NADPH oxidized/min, whereas results for the negative and positive controls were 0.02 mol of NADPH oxidized/min (Figure 8d,e).

Lipid peroxidation in *C. albicans* was evaluated by measuring the rate at which TBARS were formed; the results showed that exposure to magnetic NSS caused a gradual rise in this rate. *C. albican*’s cell counts increased by 2422 nmol/L at 0.5 MIC and 3461 nmol/L at MIC after exposure. Comparatively, the enzyme activities in the negative and positive samples were 2461 and 3730, respectively (Figure 8f). The results showed that the magnetic NSS is significant in several measures of oxidative stress in *C. albicans*.

The antioxidant system in *C. albicans* regulates antifungal resistance, morphogenesis, and immunological coping mechanisms [37]. Enzymatic and non-enzymatic scavengers, primarily CAT, SOD, GPX, GST, and GSH, constantly work to counteract ROS generation in a cell. It is believed that reactive oxygen species (ROS) play a crucial role in regulating normal metabolic processes related to survival, including cell proliferation, cell cycle progression, cell differentiation, and cell death [38]. Overexpression of the regulatory CAP1 gene in *C. albicans* mediates the production of essential antioxidant enzymes (i.e., catalase (CAT), superoxide dismutase (SOD), glutathione peroxidase (GPx), glutathione reductase (GR), and glutathione S-transferase (GST)) to repair damage caused by oxidative stress [39]. Catalase, an antioxidant enzyme that defends *C. albicans* against peroxide stress, is a member of the heme peroxidase and catalase superfamily. Catalase is responsible for reversible hydrogen peroxide (H_2_O_2_) conversion to water. After exposure to acute peroxide stress, rapid extracellular H_2_O_2_ detoxification by *C. albicans* cells primarily depends on catalase [40,41,42,43,44].

Furthermore, many glutathione peroxidases contribute to oxidative stress resistance in *C. albicans* [45]. Thus, from our results of GPx enzyme activity, it is clear that magnetic NSS can increase the activity of GPx in *C. albicans*. Consequently, the current work supports our idea that magnetic NSS induces oxidative stress in *C. albicans* via modulation of the antioxidant system, which may be complemented by cellular death and cell cycle arrest inside the cells.

### 3.6. Evaluation of Biofilm Development through MTT and Light Microscopy

The magnetic NSS showed anti-biofilm activity against *C. albicans*. At concentrations of 2XMIC and MIC displayed a percentage biofilm inhibition of 96.49% and 41.11% for *C. albicans* (Figure 9).

Furthermore, results from the crystal violet staining of the biofilms showed that an increase in the concentration of magnetic NSS led to a decrease in the density of the biofilm and a reduction in the pathogenicity, as seen by the absence of hyphae (Figure 10). The decrease in the clumping of cells illustrated by the 1/2 MIC and the increase in the number of single cells is again indicative of the effectiveness of the magnetic NSS in dissociating/disjointing the biofilm at the MIC concentration.

*C. albicans* is a prevalent human pathogen. Biofilm development is a key virulence component in *C. albicans* infections [46]. *C. albicans* biofilm is the main cause of the pathogenesis of candidiasis. It grows unchecked in the mouth, trachea, and catheters, which can lead to systemic infections; thus, there is an urgent need to find new ways to prevent and treat biofilms [29]. Drug-resistant biofilms of *C. albicans* have been found on many medical devices, which makes treatment difficult [47,48]. The antifungal and anti-biofilm activity of silver NPs have also been reported previously [49,50]. Thus, based on the results of our biofilm assay, magnetic NSS have tremendous potential for the early treatment or prevention of biofilm-related infections in immunocompromised persons.

### 3.7. The Effects on the Biofilms through Confocal Laser Scanning Microscopy

The CLSM studies further confirmed the effect of magnetic NSS on biofilm formation with confocal laser scanning microscopy. In the negative control, the biofilms made by *C. albicans* SC5314 had an unusual three-dimensional structure that was mostly made up of typical hyphal structures. The biofilm also had a clear and dense biofilm matrix that fluoresced bright green. Con A dye combines alpha-mannopyranosyl and alpha-glucopyranosyl residues present in the biofilm matrix. The Con-A enabled the visualization of healthy cells, as shown by their strong green fluorescence, whereas the FUN-1 cytosol fluorescence was red. With increased magnetic NSS concentration, biofilm inhibition was detected. Once *C. albicans* was treated with varying magnetic NSS concentrations, biofilms’ production was inhibited, as demonstrated by our data. At 1/2 of the MIC value of magnetic NSS, the biofilm lacked true hyphal structure and consisted primarily of pseudohyphae and yeast cells, whereas at MIC, the *C. albicans* biofilm exhibited an aberrant architecture primarily composed of yeast cells and lacking both true hyphal and pseudohyphal structures (Figure 11).

## 4. Conclusions

This study reported the preparation of biocompatible magnetic nanomaterials by simply co-precipitating FeCl_3_ and FeSO_4_ in the *Nigella Sativa* substrate-dispersed mixture for eco-friendly and cost-effective biomedical application. Across all assays, this composite demonstrated significant scavenging activity against free radicals, suggesting its antioxidant properties. In addition, magnetic NSS exhibits effective antifungal action against *Candida albicans*. Magnetic NSS is associated with modifying essential antioxidant enzymes, which produce oxidative stress and may result in cell cycle arrest in *C. albicans*. Furthermore, our results demonstrated that magnetic NSS have substantial anti-biofilm efficacy against *C. albicans*.

## Figures and Tables

**Figure 1 pharmaceutics-15-00642-f001:**
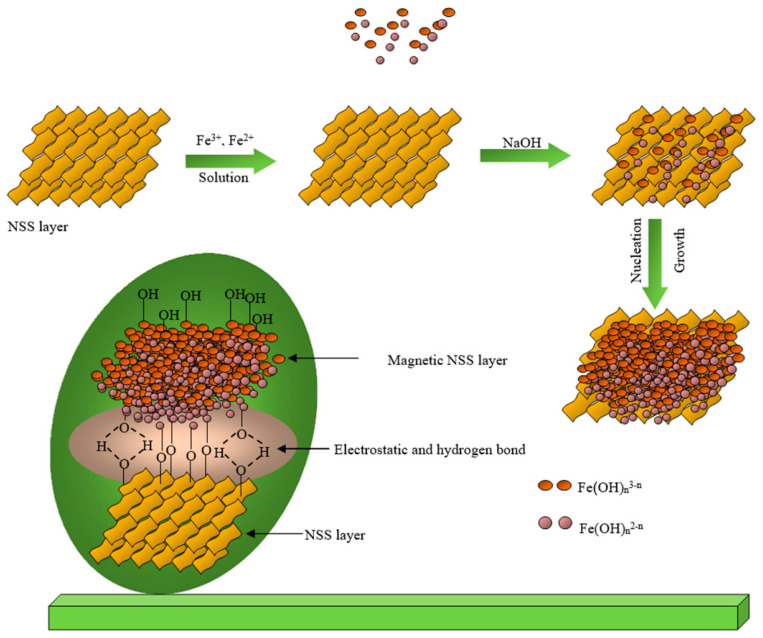
Estimated mechanism of the preparation of magnetic NSS.

**Figure 2 pharmaceutics-15-00642-f002:**
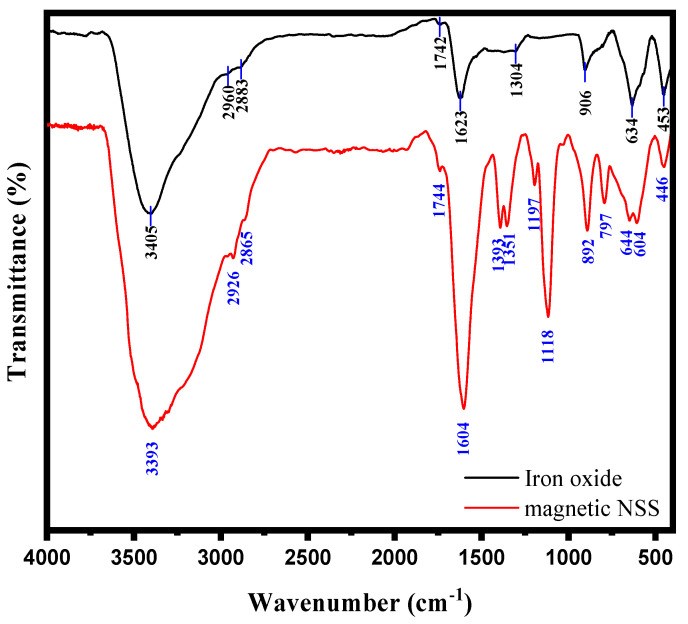
FTIR spectrum of iron oxide and magnetic NSS.

**Figure 3 pharmaceutics-15-00642-f003:**
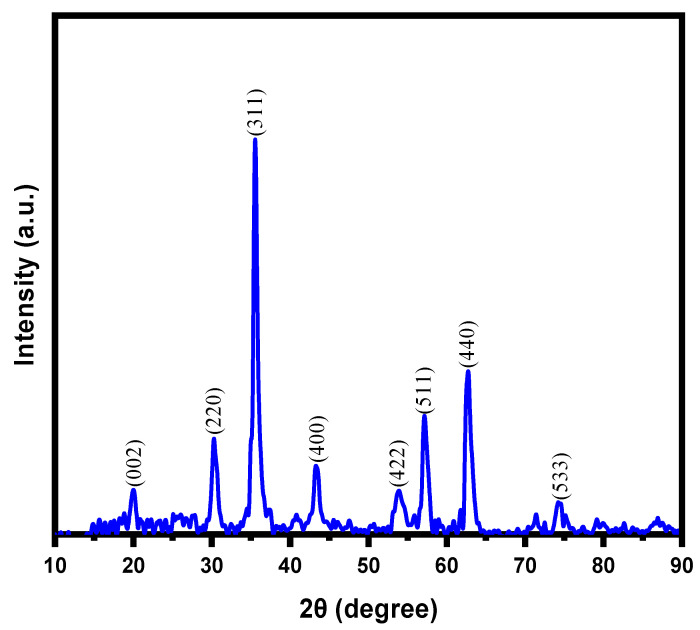
XRD pattern of magnetic NSS.

**Figure 4 pharmaceutics-15-00642-f004:**
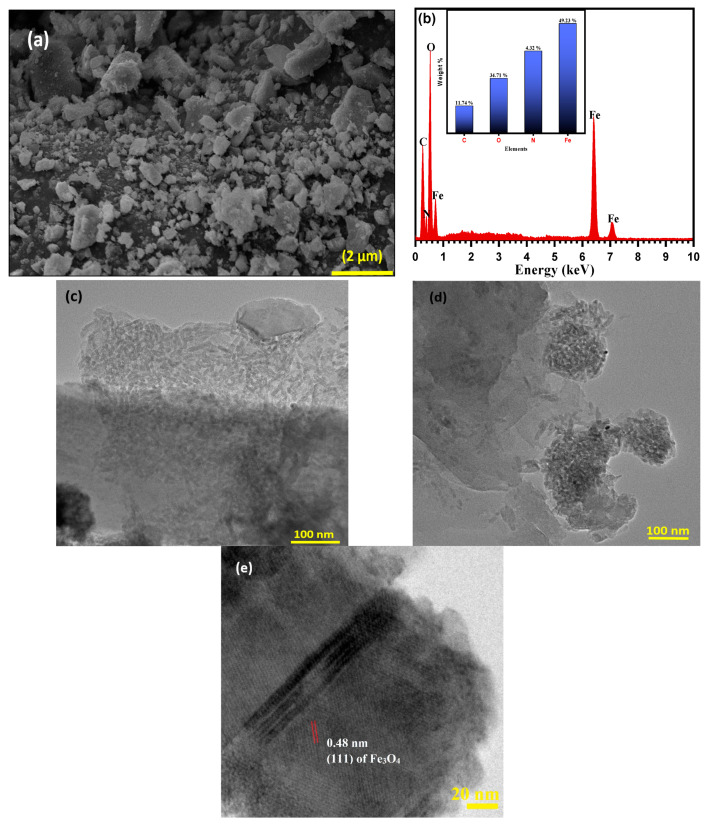
Image of (**a**) SEM of magnetic NSS, (**b**) EDX of magnetic NSS, (**c**,**d**) TEM image of magnetic NSS, and (**e**) HR-TEM image of magnetic NSS at 100 nm scale.

**Figure 5 pharmaceutics-15-00642-f005:**
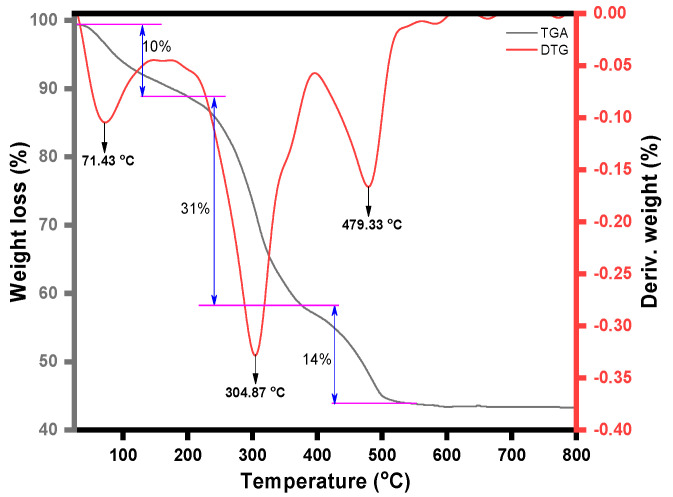
TGA and DTG graph of magnetic NSS.

**Figure 6 pharmaceutics-15-00642-f006:**
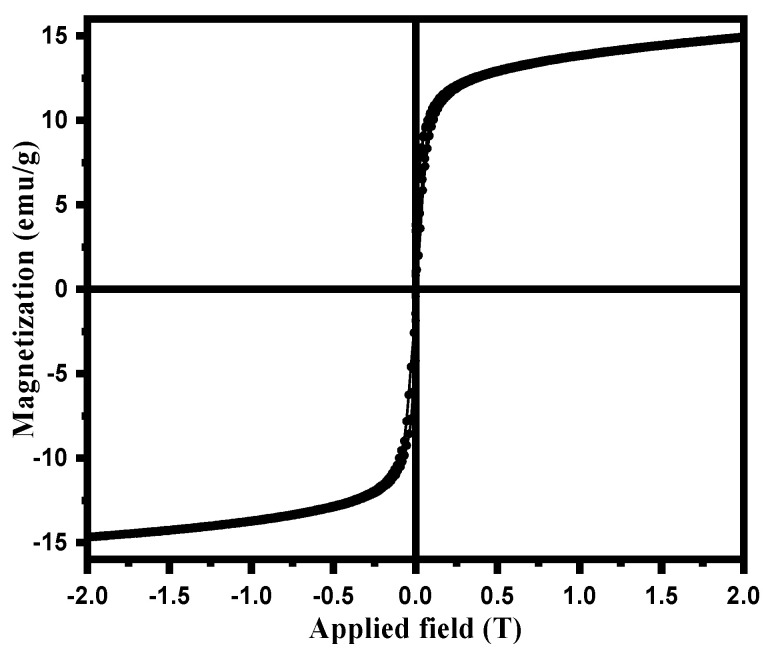
Magnetization curve of prepared magnetic NSS.

**Figure 7 pharmaceutics-15-00642-f007:**
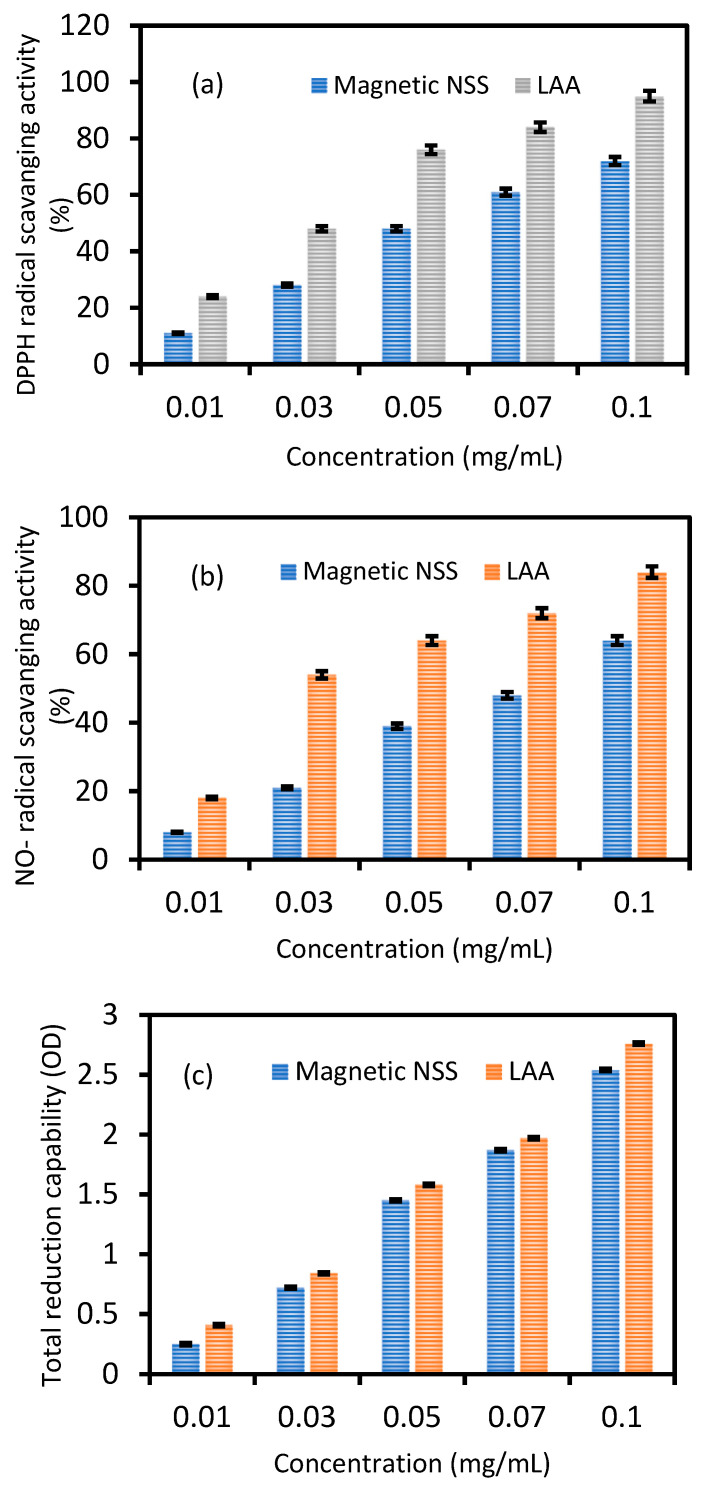
The free radical-scavenging activity of magnetic NSS on (**a**) DPPH radicals, (**b**) nitric oxide radicals, (**c**) total reduction capability. For comparison, L-AA was considered a standard antioxidant. Values are mean ± SD of three parallel measurements.

**Figure 8 pharmaceutics-15-00642-f008:**
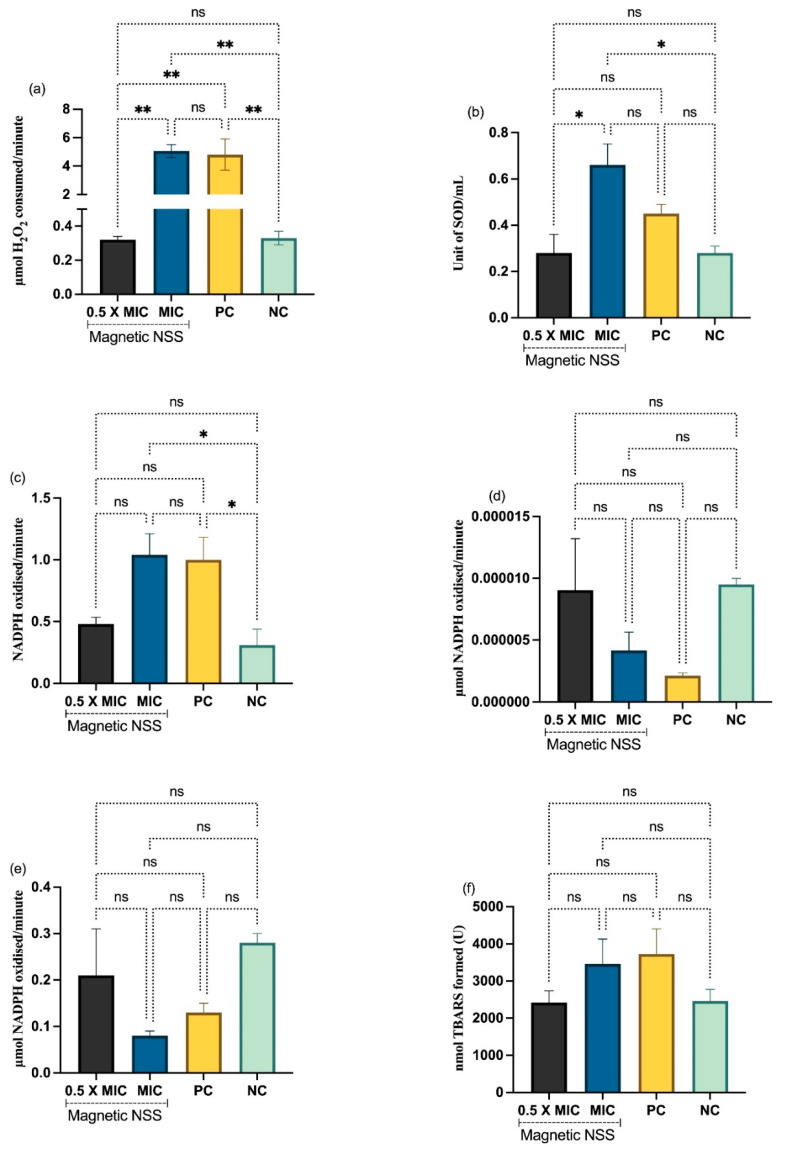
Antioxidant enzyme activity of *C. albicans*. (**a**) CAT, (**b**) SOD, (**c**) GPx, (**d**) GST, (**e**) GLR, and (**f**) TBARS. Data represent the mean of the results obtained from three independent experiments with their standard deviations as error bars. Data were analyzed using *t*-test and one-way ANOVA. ** *p* < 0.01, * *p* < 0.05 and ns indicates not significant (*p* value = 0.0606).

**Figure 9 pharmaceutics-15-00642-f009:**
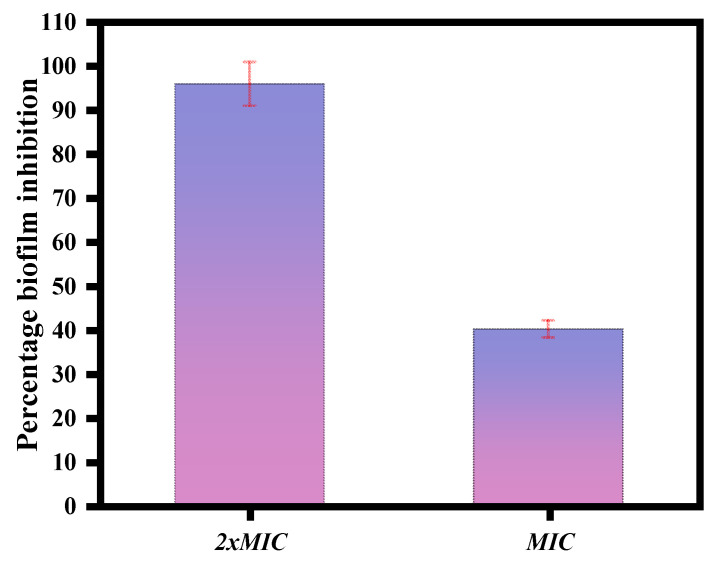
MTT assay performed on developing biofilm. Magnetic NSS was used at concentrations of 2xMIC and MIC.

**Figure 10 pharmaceutics-15-00642-f010:**
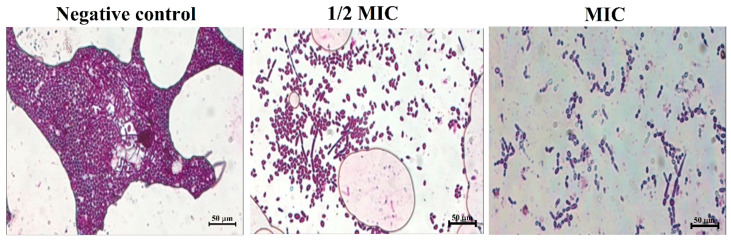
Crystal violet staining to determine the effect of magnetic NSS on developing biofilms. The negative control was left untreated and served as the growth control. The biofilms were treated with concentrations of (1/2 MIC) and (MIC). All the images were taken at 50 µm.

**Figure 11 pharmaceutics-15-00642-f011:**
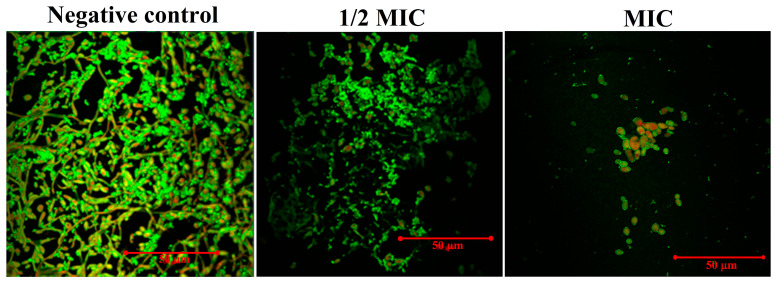
CLSM to determine the effect on developing biofilm of *C. albicans*. The negative control was left untreated and served as the growth control. The biofilms were treated with concentrations of (1/2 MIC) and (MIC). All the images were taken at 50 µm.

**Table 1 pharmaceutics-15-00642-t001:** XRD parameters.

No.	2θ (Degree)	FWHM β (°)	Crystallite Size (D) (nm)	Dislocation Density δ × 10^−3^ (nm^−2)^	Microstrain ε × 10^3^	D-Spacing (Å)	Lattice Constant (Å)
1	19.97553	0.71019	11.86309547	7.105652246	17.59607409	4.441359719	8.882719438
2	30.3536	0.8183	10.50619245	9.059605699	13.16271281	2.942340296	8.322195105
3	35.49083	0.72987	11.93954392	7.014949114	9.921778095	2.520457561	8.359412028
4	43.37817	0.86976	10.26668764	9.487227214	9.541815011	2.084315622	8.33726249
5	53.94689	1.2528	7.431478924	18.10712497	10.74066096	1.698276501	8.319821741
6	57.22792	0.80505	11.74056884	7.254737809	6.439001907	1.608458407	8.35779505
7	62.74201	0.85043	11.42716131	7.658139685	6.086027426	1.274840349	7.211586043
8	74.34727	0.84988	12.25262271	6.661037498	4.890177213	4.441359719	8.882719438

**Table 2 pharmaceutics-15-00642-t002:** MIC of NSS against *Candida albicans*.

*C. albicans* SC5314	MIC µg/mL	Fluconazole Susceptibility
NSS	3.125	Susceptible
Fluconazole	3.906	Susceptible

## Data Availability

Not applicable.

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
