# Peer review of "Facile Synthesis of Magnetic Nigella Sativa Seeds: Advances on Nano-Formulation Approaches for Delivering Antioxidants and Their Antifungal Activity against Candida albicans"

_pharmaceutics, 2023, doi:10.3390/pharmaceutics15020642_

Round 1
Reviewer 1 Report
Please check any misspelling/grammarly error.
Author Response
The manuscript was checked by English editor for the improvement of language, grammar and Typographic errors.
Reviewer 2 Report
This research is focused on the preparation of magnetic Nigella Sativa Seeds for biomedical applications such as antibiofilm, antioxidant, antifungal, etc. Authors have synthesized magnetic NSS using SEM, TEM XRD, FT-IR, VSM, and TG/DTA. overall, the results are interesting, but key issues need to solve like the raw data for the antifungal activity assay missing. The article has many grammatical and sentence errors, and the language organization needs to be improved. For these reasons, I conclude that the paper is suitable for publication with a major revision.
1. Authors may indicate the peaks using symbols for Fe3O4 and γ-Fe2O3 phases in XRD data (Figure 3)
2. In Figure 4, authors need to provide TEM images with a scale bar as well as give HR-TEM 10 nm or 5 nm scale dominating lactic for a better understanding of the morphology which is essential for medical applications.
3. Author needs to rewrite sentence “line 436-438: At every concentration that was examined, L-AA had a greater scavenging effect. In contrast to L-AA, which had an IC50 value of 0.02 0.002 mg, magnetic NSS had an IC50 value of 0.026 0.002 mg. ”
4. Authors need to change the representation of Figure 7 to a bar graph for the understanding of readers.
5. Authors have used fluconazole as Positive control in the Antifungal Activity assay. In the results section, the authors have only mentioned MIC for Magnetic NSS. Authors need to provide MIC value for Positive control fluconazole for a better understanding of the efficacy of Magnetic NSS as an antifungal agent.
6. Authors also need to provide the raw data for the antifungal activity assay and how the MIC value has been determined.
7. Typographic errors need to be corrected. The language and grammar used throughout the manuscript need to be improved
Author Response
Comments and Suggestions for Authors
This research is focused on the preparation of magnetic Nigella Sativa Seeds for biomedical applications such as antibiofilm, antioxidant, antifungal, etc. Authors have synthesized magnetic NSS using SEM, TEM XRD, FT-IR, VSM, and TG/DTA. overall, the results are interesting, but key issues need to solve like the raw data for the antifungal activity assay missing. The article has many grammatical and sentence errors, and the language organization needs to be improved. For these reasons, I conclude that the paper is suitable for publication with a major revision.
- Authors may indicate the peaks using symbols for Fe3O4 and γ-Fe2O3 phases in XRD data (Figure 3)
Response: Due to the high similarity between the XRD patterns of Fe3O4 and γ-Fe2O3, it is difficult to recognize both phases in the obtained XRD pattern [30]. Hence, the dominant phase of iron oxide in the Fe3O4/BC composite sample can be confirmed by the angular value feature of XRD data based on 2θ value of 35.35° (highest peak), corresponding to the [311] plane {Standard 2θ value for the [311] plane: 35.423° for Fe3O4 relative to 35.631° for γ-Fe2O3} [30]. Notably, the obtained 2θ value of [311] plane in this work is in good agreement (close) with the standard 2θ value for Fe3O4, indicating that Fe3O4 is the dominant phase in the prepared magnetic NSS.
- In Figure 4, authors need to provide TEM images with a scale bar as well as give HR-TEM 10 nm or 5 nm scale dominating lactic for a better understanding of the morphology which is essential for medical applications.
Response: TEM images with better resolution added in revised manuscript.
- Author needs to rewrite sentence “line 436-438: At every concentration that was examined, L-AA had a greater scavenging effect. In contrast to L-AA, which had an IC50 value of 0.02 0.002 mg, magnetic NSS had an IC50 value of 0.026 0.002 mg. ”
Response: line 436-438 rewritten (448-451)
- Authors need to change the representation of Figure 7 to a bar graph for the understanding of readers.
Response: Figure 7 changed to a bar graph, as suggested by learned reviewer.
- Authors have used fluconazole as Positive control in the Antifungal Activity assay. In the results section, the authors have only mentioned MIC for Magnetic NSS. Authors need to provide MIC value for Positive control fluconazole for a better understanding of the efficacy of Magnetic NSS as an antifungal agent.
Response: Authors thank the reviewer for the comment. As suggested the MIC value for fluconazole and the interpretation has been added to the table in results section.
- Authors also need to provide the raw data for the antifungal activity assay and how the MIC value has been determined.
Response: MIC values for both fluconazole and Magnetic NSS was determined following CLSI recommended guidelines using broth microdilution assay. There is no raw data involved as the MIC were read visually and susceptibility profiling was done as per the guidelines. For the clarification of readers, MIC determination assay was detailed in the revised methods section 2.6.
- Typographic errors need to be corrected. The language and grammar used throughout the manuscript need to be improved.
Response: We highly appreciate reviewer’s suggestions and comments that help us to improve the quality of our manuscript.
We took help of language expert to improve the language, grammar and Typographic errors.
Reviewer 3 Report
In the manuscript "Facile synthesis of magnetic Nigella Sativa seeds: Advances on nano-formulation approaches for delivering antioxidants and their Antifungal Activity against Candida albicans" the autors reported the synthesis via co-precipitation reaction of magnetic nanoparticles incorporation into natural carbon framework, derived from Nigella sativa seeds. Than, the antioxidant, antifungal and antibiofilm activities against C. albicans was evaluated.
Modification suggestions:
Nigella Sativa in Nigella sativa
Line 20: add comma after "furthermore"
Line 22: add point before "analysis"
Was the Candida albicans used in this study a clinical isolate? was it not thought to carry out a control using a collection strain?
Line 193: C. albicans in italic
Line 197: add number of reference
To evaluated the metabolic activity of biofilm, was the plate emptied and washed to remove non-adherent cells?
Line 470: C. albicans in italic
Line 581: add a comma after "further"
Author Response
Comments and Suggestions for Authors
In the manuscript "Facile synthesis of magnetic Nigella Sativa seeds: Advances on nano-formulation approaches for delivering antioxidants and their Antifungal Activity against Candida albicans" the autors reported the synthesis via co-precipitation reaction of magnetic nanoparticles incorporation into natural carbon framework, derived from Nigella sativa seeds. Than, the antioxidant, antifungal and antibiofilm activities against C. albicans was evaluated.
Modification suggestions:
Nigella Sativa in Nigella sativa
Response: Corrected as suggested by learner reviewer.
Line 20: add comma after "furthermore"
Response: Corrected.
Line 22: add point before "analysis"
Response: Corrected.
Was the Candida albicans used in this study a clinical isolate? was it not thought to carry out a control using a collection strain?
Response: In this study we used Candida albicans SC5314 which is a laboratory control strain and was previously purchased from ATCC. The strain was stored in the department at -80 ºC as glycerol stock and was revived on SD agar plates prior to the experiments. These details have now been mentioned in the revised manuscript (section 2.6).
Line 193: C. albicans in italic
Response: C. albicans has now been italicized throughout the manuscript.
Line 197: add number of reference.
Response: Reference number added.
To evaluated the metabolic activity of biofilm, was the plate emptied and washed to remove non-adherent cells?
Response: To study the metabolic activity of biofilms, all the non-adherent cells planktonic cells was aspirated with the media and by washing the plates twice with PBS. This is clearly mentioned in the methods section.
Line 470: C. albicans in italic
Response: C. albicans has now been italicized throughout the manuscript.
Line 581: add a comma after "further"
Response: Correction has been done as suggested in the revised manuscript.
Round 2
Reviewer 2 Report
Recommended for publication as the authors have addressed all the queries.